# Current Position of the Molecular Therapeutic Targets for Ovarian Clear Cell Carcinoma: A Literature Review

**DOI:** 10.3390/healthcare7030094

**Published:** 2019-07-30

**Authors:** Tsukuru Amano, Tokuhiro Chano, Fumi Yoshino, Fuminori Kimura, Takashi Murakami

**Affiliations:** 1Department of Obstetrics & Gynecology, Shiga University of Medical Science, Otsu, Shiga 520-2192, Japan; 2Department of Clinical Laboratory Medicine, Shiga University of Medical Science, Seta Tsukinowa-cho, Otsu, Shiga 520-2192, Japan

**Keywords:** ovarian clear cell carcinoma, molecular therapeutic targets, immunotherapy, anti-PD-1 antibodies, anti-PD-L1 antibodies

## Abstract

Ovarian clear cell carcinoma (OCCC) shows low sensitivity to conventional chemotherapy and has a poor prognosis, especially in advanced stages. Therefore, the development of innovative therapeutic strategies and precision medicine for the treatment of OCCC are important. Recently, several new molecular targets have been identified for OCCC, which can be broadly divided into four categories: (a) downstream pathways of receptor tyrosine kinases, (b) anti-oxidative stress molecules, (c) AT-rich interactive domain 1A-related chromatin remodeling errors, and (d) anti-programmed death ligand 1/programmed cell death 1 agents. Several inhibitors have been discovered for these targets, and the suppression of OCCC cells has been demonstrated both in vitro and in vivo. However, no single inhibitor has shown a sufficient effectiveness in clinical pilot studies. This review outlines recent progress regarding the molecular biological characteristics of OCCC to identify future directions for the development of precision medicine and combinatorial therapies to treat OCCC.

## 1. Background

Ovarian clear cell carcinoma (OCCC) accounts for 23% of all ovarian cancer cases in Japan and is much more common in Japan than ovarian carcinoma subtypes in Europe and the United States [1,2]. The histopathological features of OCCC are as follows: clear cells containing abundant glycogen, as well as hobnail cells with nuclei protrude toward the surface of cells. OCCC is closely related to endometriosis [3], and their origin is clearly distinguished from that of ovarian high-grade serous (OHGS) carcinomas, which have recently been reported to originate in the fallopian tube [4]. Gene expression profiling also revealed that OCCC is clearly distinguished from OHGS [5]. There are major molecular differences between OCCC and OHGS; for example, OCCC shows remarkably similar expression patterns to renal clear cells [6,7]. Even in clinical practice, unlike OHGS, OCCC has a low sensitivity to chemotherapy based on platinum and taxane [8,9,10]. However, OCCC patients are still treated with conventional chemotherapies such as platinum and taxane [11] because effective alternative treatments have not been identified. Therefore, the prognosis of OCCC is extremely low, particularly in advanced stages [12]. Therefore, the development of novel therapeutics for recurrent or refractory cases of OCCC is highly important. The molecular targets for OCCC can be broadly divided into the following four categories: a) downstream pathways of receptor tyrosine kinases (RTKs), b) anti-oxidative stress molecules, c) AT-rich interactive domain 1A (*ARID1A*)-related chromatin remodeling errors, and d) the programmed death ligand 1/programmed cell death 1 (PD-L1/PD-1) pathways (Figure 1). This review briefly summarizes the recent progress regarding the molecular biological characteristics of OCCC with a focus on the major candidates for targeted molecular therapy.

## 2. Downstream Pathways of RTKs

RTKs are receptors located on the surface of cells that play an important role in regulating cell proliferation, differentiation, survival, metabolism, and migration. Gain-of-function mutations have been reported in many RTKs or their downstream signaling molecules, and these mutations are thought to be associated with carcinogenesis [13]. Both the phosphoinositide 3-kinase/AKT/mammalian target of rapamycin (PI3K/AKT/mTOR) and epidermal growth factor/Ras/mitogen-activated protein kinase (EGF/Ras/MAPK) pathways are downstream pathways of RTKs; several genetic mutations and the amplification of key components related to these pathways have been reported in OCCC. Specifically, phosphatidylinositol-4,5-bisphosphate 3-kinase catalytic subunit alpha (*PIK3CA*) mutations; the loss of phosphatase and tensin homolog (PTEN) expression; and the amplification of human epidermal growth factor receptor 2 (HER2), MET (also known as hepatocyte growth factor receptor; HGFR), and ADP-ribosylation factor-like 4C (ARL4C) expression have been shown to activate these pathways in OCCC (Figure 2). 

### 2.1. MET/HGFR

MET, also known as HGFR, is an RTK encoded by the *MET* gene. MET relays signals through both the MAPK and PI3K pathways and plays an important role in promoting proliferation and invasion in various tumors. MET amplification was observed in gefitinib-resistant lung cancer cases, suggesting a relationship between MET and drug resistance [14]. In OCCC, MET amplification was observed in 37% of cases and was associated with a poor prognosis [15]. Recent research has revealed that MET inhibitors significantly decreased the proliferation and increased the apoptosis of OCCC cells in vitro and also suppressed in vivo tumor growth in xenograft models of OCCC [16]. However, the MET inhibitor cabozantinib was clinically ineffective in thirteen patients with recurrent OCCC, [17]. In contrast, this drug showed therapeutic excellency for clear cell renal carcinoma (CCRC), holding quite similarly pathological phenotype. Panagiotis et al. advocated that the differences in the treatment outcomes between OCCC and CCRC might be due to the prevalence of von Hippel-Lindau mutations in CCRC as opposed to the absence of these mutations in OCCC, as well as to the increased incidence of active abnormalities of the PI3K/mTOR/AKT pathway and of ARID1A mutations in OCCC [17].

### 2.2. HER2

HER2 is another RTK involved in the regulation of cell functions such as proliferation, differentiation, migration, and survival protein expression. Through immunohistochemistry (IHC), HER2 overexpression was observed in 42% of OCCC cases, and at levels twofold higher than those of OHGS carcinoma [18]. However, the poor response rate of the single agent trastuzumab, a monoclonal humanized anti-HER2 antibody was reported in 2003 in recurrent ovarian cancers showing HER2 overexpression (forty-one recurrent ovarian cancer cases, including seven clear cell types) [19]. The discordance between immunological evaluation and therapeutic responsiveness may be caused by the difference of applied antibodies to each procedure. Additionally, Koopman et al. demonstrated that in OCCC, there is a considerable difference in HER2 overexpression evaluated by different IHC antibodies and an apparent discordance with in-situ hybridization [20]. In the future era of precision medicine, the global RNA transcript profile should be additionally considered for evaluation of HER2 activation in OCCC treatment, together with each cancer DNA mutation and protein status.

### 2.3. PIK3CA Mutations

The *PIK3CA* gene, encoding a catalytic subunit of PI3K, is among the most important genetic abnormalities related to the PI3K/AKT/mTOR pathway. Active *PIK3CA* mutations in OCCC were first reported by Kuo et al. [21]. The incidence of *PIK3CA* mutations in OCCC has been reported in about 30–40% of cases [21,22,23], and these mutations are thought to occur as early events in OCCC tumorigenesis [22]. PI3Ks are lipid kinases that phosphorylate phosphatidylinositol to form phosphatidylinositol (3,4,5)-trisphosphate (PIP3). PIP3 regulates various cell processes related to cancer development, including cell proliferation, survival, and apoptosis through the activation of PI3K/AKT/mTOR signaling. *PIK3CA* mutation or amplification was reported to be associated with an improved prognosis for OCCC patients; however, these mutations did not affect the sensitivity of PI3K/AKT/mTOR inhibitors in vitro [24].

### 2.4. PTEN Loss

PTEN, whose gene is located on chromosome 10q23 in humans, is a phosphatase enzyme that dephosphorylates PIP3 and has the reverse function of PI3K [25]. The loss of PTEN activity, which activates the PI3K/AKT cascade, has been reported in 27.5–37.5% of OCCC cases [26,27]. As mutations in the *PTEN* gene have been observed in only 5% of OCCC cases [21], it is possible that epigenetic regulation plays a central role in the loss of PTEN activity. Sato et al. [28] reported that the inactivation of PTEN is an early event in the development of OCCC from endometriosis. The relationship between PTEN loss and the prognosis of OCCC or the sensitivity of PI3K/AKT/mTOR inhibitors remain unknown.

### 2.5. ARL4C

ARL4C is a small GTP-binding protein (small G protein) that is widely expressed in the human body, including in the ovarian follicular cells. The simultaneous activation of both the Wnt/β-catenin and EGF/Ras/MAPK pathways induces ARL4C expression, which results in epithelial tubular morphogenesis and growth, promoting the tubular formation of the epithelium [29,30]. Recent research has revealed that ARL4C is frequently overexpressed in colorectal and lung cancers and that it plays an important role in tumor progression by promoting the proliferation, migration, and invasion of cancer cells [31]. It has also been demonstrated that the overexpression of ARL4C was detected in about 60% (25 out of 41) of OCCC cases, and that ARL4C predicted a poor prognosis in endometriosis-associated ovarian cancers, including OCCC [32]. The activation of small G proteins, including ARL4C, requires the binding of Cys residues to the cell membrane by C-terminal prenylation. ARL4C contains Cys residues on its C-terminal region, and the prenylation of these residues is essential for ARL4C activation [33]. As statins and bisphosphonates are major candidates for inhibiting the mevalonate pathway and for decreasing the downstream prenylation of substrates, drug repositioning targeting ARL4C with statins and bisphosphonates may be useful against ARL4C overexpression in OCCC, at least during the post-operative follow-up periods, when it is not possible to clinically detect any apparent tumor mass.

### 2.6. Clinical Use of Rtk/pi3k/Akt/Mtor Inhibitors for OCCC

Several clinical trials have been conducted investigating the effectiveness of PI3K/AKT/mTOR inhibitors for treating OCCC. Takano et al. [34] reported that the clinical use of temsirolimus, an mTOR inhibitor, resulted in one partial response and one stable disease out of six recurrent chemotherapy-resistant OCCC patients. The combination of temsirolimus with carboplatin/paclitaxel was investigated in patients with advanced OCCC. However, compared to conventional treatments, this regimen did not significantly increase the rate of progression-free survival [35]. Thus, despite its effectiveness in in vitro experiments, no clinical advantages have been observed for PI3K/AKT/mTOR inhibitors in treating OCCC. Further investigations are required to determine which drugs are effective in combination with PI3K/AKT/mTOR inhibitors, or which mutations associated with OCCC will be affected by PI3K/AKT/mTOR inhibitor treatment.

## 3. Anti-Oxidative Stress

Oxidative stress induced by excess heme production and iron accumulation was revealed to be an important trigger in the malignant transformation of endometriosis to OCCC [36]. OCCC is regarded as an oxidant stress-tolerant cancer because it arises from chocolate cysts in a highly oxidative microenvironment induced by excess heme production and chronic inflammation. In OCCC, abnormalities are often found in genes with roles in the oxidative stress response and the metabolism of reactive oxygen species (ROS) [37]. The overexpression of hepatocyte nuclear factor 1 homeobox B (HNF1B), a major homeobox-containing protein also known as transcription factor-2 (TCF2); its overexpression contributes to cell survival under stresses such as chemo-reagents. HNF1B can modify and adapt cancer cells to survive under hypoxia and acidosis in a process facilitated by increased glucose consumption and glycolysis, commonly known as the Warburg effect [38]. The overexpression of superoxide dismutase 2 (SOD2), an antioxidant enzyme, can also advance tumor growth and metastasis in OCCC (Figure 3) [39].

### 3.1. HNF1B

The overexpression of HNF1B is known to contribute to cell survival under stresses such as hypoxia, acidosis, nutrient starvation, and chemotherapeutic reagents by promoting glycogen metabolism pathways such as gluconeogenesis, glycogen accumulation, and aerobic glycolysis. Tsuchiya et al. [40] first reported HNF1B overexpression in OCCC, and that reduced HNF1B expression considerably increased the rates of apoptosis in two OCCC cell lines. Similarly to the mutation of *ARID1A* (discussed below, Section 4), the overexpression of HNF1B was observed in endometrial tissues adjacent to OCCC tumors, suggesting that HNF-1B overexpression occurs as an early event of OCCC carcinogenesis. According to previous studies, HNF1B overexpression in OCCC is caused by epigenetic changes rather than by mutations. Kato et al. [41] revealed that the hypomethylation of the CpG island of *HNF1B* induced its overexpression in OCCC. Previous studies have revealed the roles of HNF1B in OCCC in driving the expression of several characteristic genes associated with OCCC [42]: stimulating changes in OCCC metabolism to promote gluconeogenesis, glycogen accumulation, and aerobic glycolysis [43], inducing chemotherapeutic resistance through the suppression of sulfatase-1 (Sulf-1), an extracellular sulfatase catalyzing the 6-O desulfation of heparan sulfate glycosaminoglycans. [44], and reducing the activity of immunological checkpoints against tumors. Moreover, recent research has revealed that expression HNF1B high promotes dedifferentiation to cancer stem cells via activation of the Notch pathway, and that HNF1B enhances invasive potential and the epithelial-mesenchymal transition (EMT) in cancer cells [45]. Thus, HNF1B is a major potential target for the treatment of OCCC. However, effective inhibitors have not been identified.

### 3.2. SOD2

Mitochondrial SOD2 is an enzyme that metabolizes superoxide in mitochondria and plays an important role in maintaining mitochondrial function through oxidative stress tolerance. Hemachandra et al. [39] revealed that SOD2 was more highly expressed in OCCC than in any other epithelial ovarian cancer subtypes and that its overexpression contributed to tumor growth and metastasis in a chorioallantoic membrane (CAM) model. They also indicated that its expression was associated with increased cell proliferation, migration, outgrowth on collagen, spheroid attachment, and Akt phosphorylation in ES-2 OCCC cells [39]. Therefore, SOD2 is regarded as a pro-tumorigenic or metastatic factor in OCCC. Clinical research has recently demonstrated that high SOD2 expression is observed in 76% (33 out of 41) of OCCC cases, and that SOD2 overexpression is correlated with a poor prognosis for OCCC [46]. The above studies support the model that OCCC affects mitochondrial function through SOD2 overexpression, which is therefore a potential therapeutic target. Replacement therapy or drug repurposing targeting tumor cell mitochondria, which can be achieved using biguanides, an agent used to treat diabetes mellitus, may improve the therapeutic effect of OCCC treatment.

## 4. *ARID1A* Chromatin Remodeling Abnormalities

The *ARID1A* gene encodes BAF250a, a subunit of the switch/sucrose nonfermentable (SWI/SNF) chromatin-remodeling complex. The SWI/SNF complex modifies chromatin structure by histone octamer ejection, octamer sliding, or local chromatin unwrapping to allow the binding of other transcription factors [47]. Therefore, the SWI/SNF complex is regarded a master regulator of transcription. Recent studies have revealed that this complex contributes to transcriptional repression through c-Myc inhibition, and to the prevention of DNA entanglement during mitosis by inducing topoisomerase IIα (TOP2A) expression [48,49]. Furthermore, *ARID1A* mutations are correlated with *PIK3CA* mutations, which causes the phosphorylation of multiple PI3K members, leading to the activation of the PI3K/AKT/mTOR pathway (Figure 4) [50]. These observations suggest that *ARID1A* acts as a tumor suppressor gene. In fact, *ARID1A* mutations are frequently found in several carcinomas, such as stomach cancer and bladder cancer and are observed in in 46–60% of OCCC cases [47,51]. Because inactivating *ARID1A* mutations and the loss of BAF250a function have been identified in atypical endometriosis at the periphery of OCCC tumors, they are believed to be majorly involved in the development of OCCC from endometriosis [51]. Thus, inactivating mutations of *ARID1A* can contribute to OCCC carcinogenesis through chromatin remodeling errors and the activation of the PI3K pathway.

Although several studies investigating the relationship between *ARID1A* mutation and prognosis in OCCC have been conducted, the correlation is still unclear [52,53,54]. Recently, attempts to target *ARID1A* mutations for OCCC treatment have been conducted. Bitler et al. [55,56] reported that inhibiting the enhancer of zeste homolog 2 (EZH2) histone methyltransferase activity induced synthetic lethality in *ARID1A*-mutated OCCC tumors via PI3K/AKT signaling. Highly specific small-molecule EZH2 inhibitors have already been developed and have entered clinical trials for hematopoietic malignancies such as diffuse large B cell lymphoma. Clinical trials investigating their effectiveness in treating OCCC are therefore required. Additionally, Berns et al. [57] recently revealed that small-molecule inhibitors of the bromodomain and extra-terminal domain (BET) family of proteins inhibit the proliferation of *ARID1A*-mutated OCCC cells by reducing the expression of multiple SWI/SNF members both in vitro and in vivo. EZH2 and BET inhibitors may represent promising new drugs to treat *ARID1A*-mutated OCCC in the future.

## 5. Anti-PD-1/PD-L1 Agents

Immune checkpoint blockade therapeutics have become more ubiquitous due to their ease of administration, favorable side effect profile, and effectiveness in certain tumor types. Given the success of checkpoint inhibitors in the treatment of other malignancies, there has been an attempt to replicate these results in ovarian cancer clinical trials. A clinical phase II trial, 2 out of 20 platinum-resistant recurrent OCCC cases showed complete remission upon treatment with an anti-PD-1 antibody [58,59]. Recently, Mismatch repair (MMR) deficiencies have attracted attention as a predictive marker for this type of immunotherapy. Several works revealed that MMR-deficient tumors are sensitive to immune checkpoint blockade by anti- PD-1 or anti-PD-L1 antibody [60,61]. Only a small proportion of ovarian cancers are thought to be due to germ-line *MMR* mutations [62] and MSI-H ovarian cancers are rare. Recent research demonstrated that about 7% (4 out of 57) of OCCC cases had MSI-H cancers without any *MMR* mutations [63]. While the applied population is small, immunotherapy by anti-PD-1/PD-L1 antibodies shows a high potential as an effective treatment strategy for MSI-H OCCC. On the other hand, two OCCC patients who could achieve complete remission showed microsatellite stability (MSS) in the clinical trial mentioned above [58,59]. Taken together, in OCCC it is clear that not only *MMR* status, but also various genetic or epigenetic modifications may contribute to the sensitivity of immune checkpoint blockade, whose predictive efficiency should be precisely clarified in subsequent studies.

## 6. Conclusions

In this review, we briefly summarized recent discoveries regarding the biological molecular characteristics of OCCC with a focus on molecular candidates for therapeutic targets. The genetic and biological characteristics of OCCC are slowly being clarified, and the therapeutic effects of various anti-cancer drugs, molecular targeting drugs, drug repositioning strategies, and immunotherapies have been verified. Despite this, a low proportion of patients who are highly sensitive to immunotherapy by anti-PD-1 or anti-PD-L1 antibodies have received significant benefits from the treatment. Thus, not only further investigations to identify novel molecular targets, but also personalized medicine combining multiple treatments based on the genetic and molecular characteristics of individual tumors, are desired. On the other hand, it has been challenging to develop small-molecule inhibitors for molecules such as HNF1B. Therefore, we believe that it is absolutely essential for innovative treatments against OCCC to introduce nucleic acid-based medicine and combinatorial treatment to adapt to the transcriptional profile of each individual tumor.

## Figures and Tables

**Figure 1 healthcare-07-00094-f001:**
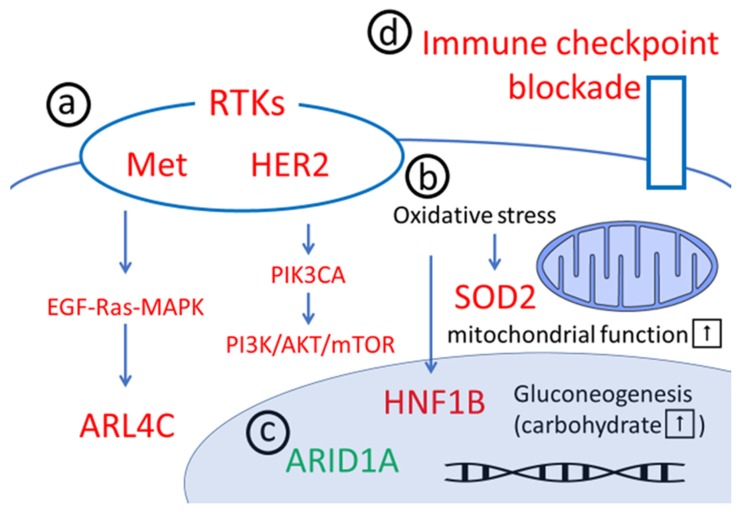
The four major categories of candidate molecular targets for the treatment of ovarian clear cell carcinoma: (**a**) downstream pathways of receptor tyrosine kinases (RTKs), (**b**) the anti-oxidative stress pathway, (**c**) AT-rich interactive domain 1A (ARID1A) chromatin remodeling disorders, and (**d**) anti-programmed death ligand 1/programmed cell death 1 (PD-L1/PD-1) agents. Red indicates members playing a pro-oncogenic role. Green represents molecules with tumor suppression functions. ARL4C: ADP-ribosylation factor-like 4C, EGF: epidermal growth factor, HER2: human epidermal growth factor receptor 2, HNF1B: hepatocyte nuclear factor 1 homeobox B, MAPK: mitogen-activated protein kinase, mTOR: mammalian target of rapamycin, PIK3CA: phosphatidylinositol-4,5-bisphosphate 3-kinase catalytic subunit alpha, PI3K: phosphoinositide 3-kinase, SOD2: superoxide dismutase 2.

**Figure 2 healthcare-07-00094-f002:**
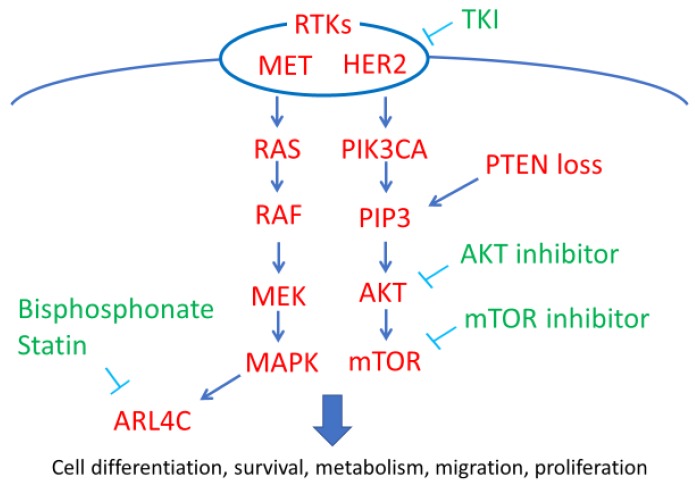
Receptor tyrosine kinases (RTKs) and their downstream pathway members in ovarian clear cell carcinoma (OCCC). RTK and its downstream pathways (red) play a pro-oncogenic role in OCCC. Tyrosine kinase inhibitors (TKI), AKT inhibitors, mammalian target of rapamycin (mTOR) inhibitors, and bisphosphonate (green) show potential for use as new therapeutic strategies. ARL4C: ADP-ribosylation factor-like 4C, HER2: human epidermal growth factor receptor 2, MAPK: mitogen-activated protein kinase, MEK: MAPK kinase, PIK3CA: phosphatidylinositol-4,5-bisphosphate 3-kinase catalytic subunit alpha, PIP3: phosphatidylinositol (3,4,5)-trisphosphate.

**Figure 3 healthcare-07-00094-f003:**
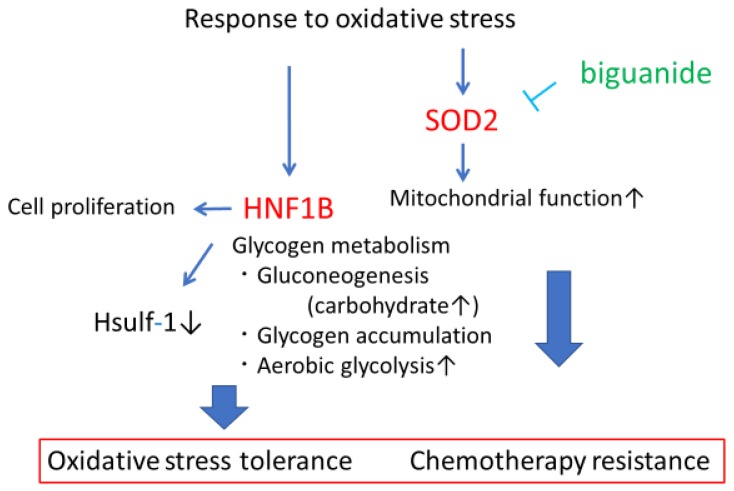
The role of anti-oxidative stress molecules in ovarian clear cell carcinoma (OCCC) progression. Hepatocyte nuclear factor 1 homeobox B (HNF1B) and superoxide dismutase 2 (SOD2) (red) play an important role in cell survival under oxidative stress in OCCC. Oxidative stress and the oxidative stress response play a major role in the overexpression of HNF1B and SOD2 in OCCC. Effective small-molecule inhibitors of HNF1B have not been developed. Drug repurposing by biguanide (green) may be effective for SOD2-overexpressing OCCC cases. Sulf-1: sulfatase-1.

**Figure 4 healthcare-07-00094-f004:**
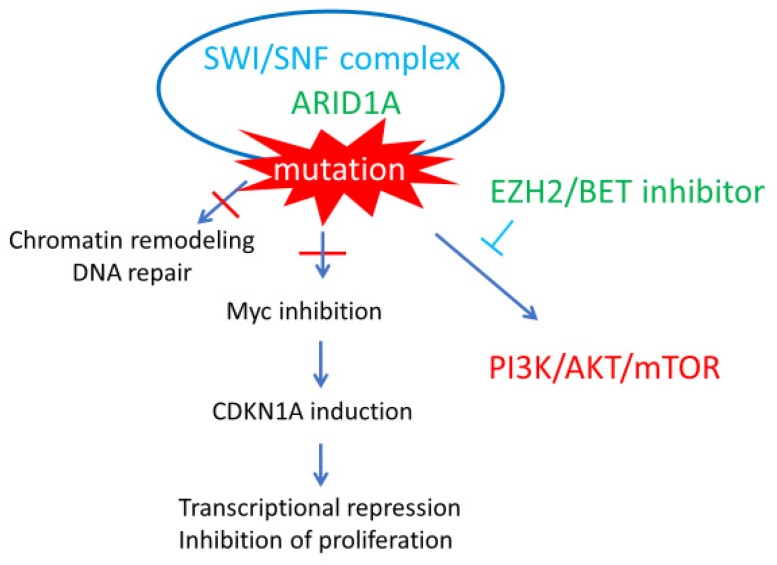
AT-rich interactive domain 1A (*ARID1A*) chromatin remodeling abnormalities in ovarian clear cell carcinoma (OCCC). ARID1A (green), a member of the switch/sucrose nonfermentable (SWI/SNF) complex (blue), works as a tumor suppressor through chromatin remodeling, DNA repair, and Myc inhibition. It also plays a role in the phosphoinositide 3-kinase/AKT/mammalian target of rapamycin (PI3K/AKT/mTOR) pathway (red). *ARID1A* mutations are characteristic and frequent in OCCC, indicating its potential as an effective therapeutic target. Inhibitors of enhancer of zeste homolog 2 (EZH2) and bromodomain and extra-terminal domain protein (BET) induced synthetic lethality in ARID1A-mutated OCCC tumors via PI3K/AKT signaling (green). CDK1A: cyclin-dependent kinase inhibitor 1A.

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
