# Peer review of "Current Position of the Molecular Therapeutic Targets for Ovarian Clear Cell Carcinoma: A Literature Review"

_healthcare, 2019, doi:10.3390/healthcare7030094_

Round 1

Reviewer 1 Report

I reviewed the manuscript by Amano and colleagues titled "Current position of the molecular therapeutic targets for ovarian clear cell carcinoma: a literature review". The reviewers summarize the molecular pathways implicated in ovarian cancer clear cell carcinoma and potential therapeutic implications. I have the following comments:

The authors should elaborate more on the therapeutic targets and the current status of these targets in preclinical models and clinical trials.

In Figure 1 and Figure 3, carbon hydrate should be carbohydrate.

Figure 3 legend, "repositioning" should be repurposing. 

The manuscript needs English editing.

Author Response

To Reviewer 1

Re: Mamuscript #healthcare-556187

Title: Current position of the molecular therapeutic targets for ovarian clear cell carcinoma: a literature review

Thank you for your constructive comments. The comments raised were given attention and the manuscript was revised. The corrected parts are in blue letters. We give our permission for any language corrections you intend to make.

1)     In Figure 1 and Figure 3, carbon hydrate should be carbohydrate.

We are sorry for the mistake. We have corrected the word to “carbohydrate”.

2)     Figure 3 legend, "repositioning" should be repurposing.

Thank you for your advice. We agree with you. We have corrected the word in Figure3 legend and section3.2 (Page 6, Line 210) “repositioning” to “repurposing”.

3)     The manuscript needs English editing.

Thank you for your advice. We requested Editage(www.editage.jp) for the English language review following your advice. This paper got English proofreading by Editage .

Your kind consideration of our manuscript would be greatly appreciated. We hope you will find this version acceptable for publication in “Healthcare”.

Reviewer 2 Report

The manuscript entitled “Current position of the molecular therapeutic targets for ovarian clear cell carcinoma: a literature review” by Amano et al. described the current state of chemotherapeutic strategies for ovarian clear cell carcinoma (OCCC). The authors roughly categorized the chemotherapeutic agents for OCCC into four groups and discussed gene by gene.

 The manuscript is clearly written and easy to be read. However, the discussions for each gene may not be enough and the some of the essential papers are not cited whereas some of the minor therapeutic targets such as ARL4C has been described pretty much. The reviewer recommends to cite some critical papers to fortify their well-written review.

1.       They need to discuss the reason why the MET-inhibitors were not effective on OCCC although the OCCC expression profile is similar to the renal clear cell carcinoma. According to the Panagiotis et al, (Gynecol Oncol, 2018), a MET inhibitor Cabozantinib did not work for OCCC although the drug showed excellent results for renal clear cell carcinoma. The authors should cite this paper and discuss the possible reasons of the difference of effects of cabozantinib on OCCC and on renal clear cell carcinoma.

2.       In the case of HER2, the estimate of overexpression of the gene may be unstable. Koopman et al. (Histopathology 2018) describes the discordance of HER2 expression estimates among the antibodies or in-situ hybridization. And the ratio may not be so high as the authors statement in this manuscript. The authors should comment on the importance of evaluation of HER2 expression in OCCC.

3.       In terms of microsatellite instability, as authors stated, the immune checkpoint therapy is very effective regardless to the stability of microsatellite. So, the reviewer recommends to change the headline as “Anti-PD-1/PD-L1 agents” and rewrite the section accordingly.

Author Response

To Reviewer 2

Re: Mamuscript #healthcare-556187

Title: Current position of the molecular therapeutic targets for ovarian clear cell carcinoma: a literature review

Thank you for your constructive comments. The comments raised were given attention and the manuscript was revised. The corrected parts are in blue letters. We give our permission for any language corrections you intend to make.

1)     They need to discuss the reason why the MET-inhibitors were not effective on OCCC although the OCCC expression profile is similar to the renal clear cell carcinoma. According to the Panagiotis et al, (Gynecol Oncol, 2018), a MET inhibitor Cabozantinib did not work for OCCC although the drug showed excellent results for renal clear cell carcinoma. The authors should cite this paper and discuss the possible reasons of the difference of effects of cabozantinib on OCCC and on renal clear cell carcinoma.

Following your fruitful comments, we have modified the sentences as follows in the section2.1 (Page 3, Line 90-95):

In contrast, this drug showed therapeutic excellency for clear cell renal carcinoma (CCRC), holding quite similarly pathological phenotype. Panagiotis et al. advocated that the differences in the treatment outcomes between OCCC and CCRC might due to the prevalence of von Hippel-Lindau mutations in CCRC as opposed to the absence of these mutations in OCCC, and also due to the increased incidence of active abnormalities of the PI3K/mTOR/AKT pathway and of ARID1A mutations in OCCC [19].

2)     In the case of HER2, the estimate of overexpression of the gene may be unstable. Koopman et al. (Histopathology 2018) describes the discordance of HER2 expression estimates among the antibodies or in-situ hybridization. And the ratio may not be so high as the authors statement in this manuscript. The authors should comment on the importance of evaluation of HER2 expression in OCCC.

Thank you for your suggestion. We have modified the sentence (Page3, Line 103-108) as follow:

The discordance between immunological evaluation and therapeutic responsiveness may be caused by the difference of applied antibodies to each procedure. Additionally, Koopman et al. have demonstrated that in OCCC, there is considerable difference in HER2 overexpression evaluated by different IHC antibodies, and is apparent discordance with insitu hybridization [20]. From now on precision medicine era, together with each cancer DNA mutation and protein status, the RNA transcript profile should be considered for evaluation and treatment of HER2 activation in OCCC.”

3)     In terms of microsatellite instability, as authors stated, the immune checkpoint therapy is very effective regardless to the stability of microsatellite. So, the reviewer recommends to change the headline as “Anti-PD-1/PD-L1 agents” and rewrite the section accordingly.

Thank you for your advice. We agree with you. We changed the headline of section 5 as “Anti-PD-1/PD-L1 agents, modified the abstract section (Page 1, Line 17-18), the background section (Page 2, Line 53) and Figure 1, and rewrote the section as follows (Page 7, Line 253- Page 8, Line 270) :

Immune checkpoint blockade therapeutics have become more ubiquitous due to their ease of administration, favorable side effect profile, and effectiveness in certain tumor types. Given the success of checkpoint inhibitors in the treatment of other malignancies, there has been an attempt to replicate these results in ovarian cancer clinical trials. In a clinical phase II trial, 2 out of 20 platinum-resistant recurrent OCCC cases showed complete remission upon treatment with an anti-PD-1 antibody [59, 60]. Recently, Mismatch repair (MMR) deficiencies has attracted attention as a predictive marker for this type of immunotherapy. Several works have revealed that MMR-deficient tumors are sensitive to immune checkpoint blockade by anti- PD-1 or anti-PD-L1 antibody [61,62]. Only a small proportion of ovarian cancers are thought to be due to germ-line MMR mutations [63], and MSI-H ovarian cancers are rare. Recent research demonstrated that about 7% (4 out of 57) of OCCC cases had MSI-H cancers without any MMR mutations [64].  While the applied population is small, immunotherapy by anti-PD-1/PD-L1 antibodies shows high potential as an effective treatment strategy for MSI-H OCCC. On the other hand, two OCCC patients who could achieve complete remission showed microsatellite stability (MSS) in the clinical trial mentioned above [62,63]. Taken together, in OCCC it is clear that not only MMR status, but also various genetic or epigenetic modifications, may contribute to the sensitivity of immune checkpoint blockade, whose predictive efficiency should be precisely clarified in subsequent studies.” 

Your kind consideration of our manuscript would be greatly appreciated. We hope you will find this version acceptable for publication in “Healthcare”.